# Evaluating a Novel Hydrocolloid Alternative for Yogurt Production: Rheological, Microstructural, and Sensory Properties

**DOI:** 10.3390/foods14132252

**Published:** 2025-06-25

**Authors:** F. N. U. Akshit, Ting Mao, Shwetha Poojary, Venkata Chelikani, Maneesha S. Mohan

**Affiliations:** 1Alfred Dairy Science Laboratory, Department of Dairy and Food Science, South Dakota State University, Brookings, SD 57006, USA; akshitbastali@gmail.com (F.N.U.A.); ting.mao@sdstate.edu (T.M.); 2Department of Wine, Food and Molecular Biosciences, Lincoln University, Lincoln 7647, New Zealand; shwethapoojary3@gmail.com (S.P.); venkata.chelikani@lincoln.ac.nz (V.C.)

**Keywords:** hydrocolloids, viscosity, disodium 5-guanylate, syneresis, microstructure

## Abstract

This study aimed to assess the viability of a new gelling agent, formed by a combination of disodium 5-guanylate and lactic acid, as a potential substitute for conventional hydrocolloids in yogurt production. Six different yogurt samples containing novel gel (combination of lactic acid and disodium 5-guanylate), disodium 5-guanylate, gelatin, agar-agar, lactic acid, and a control yogurt without any hydrocolloid or other additives, were studied. As expected, all the yogurt samples exhibited shear-thinning behavior. The novel gel yogurt, when compared to the control yogurt, displayed similar viscosity at a low shear rate of 4.5 s^−1^ (mimicking the shearing during manual scooping with a spoon) and lower viscosity at a shear rate of 60.8 s^−1^ (mimicking the agitation in the mouth). Notably, the novel gel yogurt demonstrated a lower flow behavior index (0.13 vs. 0.40 on day 1), reduced syneresis (23.37% vs. 33.75%), and had a higher consistency coefficient (9.2 vs. 7.25 on day 1) compared to the control yogurt. The novel gel yogurt exhibited superior rupture strength compared to yogurt with other hydrocolloids, such as gelatin and agar-agar, and similar brittleness to yogurt with gelatin. Microstructural analysis revealed an aggregated and compact protein network in the novel gel yogurt, analogous to the yogurt with gelatin. Sensory evaluations indicated no significant differences between the control and the novel gel yogurt. Therefore, the novel gelling agent studied can serve as a cost-effective alternative in yogurt production, compared to conventional hydrocolloids that are in short supply, in high demand, and expensive in the market.

## 1. Introduction

Yogurt is a fermented milk product, wherein the production of lactic acid by yogurt starter culture (*Streptococcus thermophilus* and *Lactobacillus delbrueckii subsp. bulgaricus*) reduces the pH slowly to the isoelectric point of casein, leading to the formation of a gel network. Yogurt is one of the most consumed dairy products worldwide due to its sensory, nutritional, and health-beneficial properties [1]. The current market revenue for yogurt in the United States is USD 11.66 billion and is projected to grow at a 4.49% compounded annual growth rate, reaching USD 13.9 billion by 2028 [2].

The textural attributes of yogurt are as crucial as its taste and aroma [3]. Yogurt manufacturers face two major texture-related challenges: changes in viscosity and syneresis during storage [3,4,5]. These defects arise from diverse factors such as variations in milk composition, ingredient type and quantity, processing methods, incubation, and storage conditions [6]. To maintain the quality of yogurt throughout its shelf life, dairy ingredients (sodium caseinate, whey protein concentrates, skim milk powder, and buttermilk powder) and hydrocolloids (gelatin, starch, carrageenan, xanthan gum, guar gum, and locust bean gum) are incorporated into yogurt formulations [3,5,7]. Hydrocolloids are polysaccharide or protein-based long-chain biopolymers that are easily dispersible, can help mitigate syneresis, and induce changes in viscosity during storage [4]. Gelatin stands out as the predominant hydrocolloid in yogurt formulation, with its ability to melt at body temperature and provide yogurt with desirable texture and sensory attributes [8]. Although gelatin is an effective gelling agent, its animal origin (bovine or porcine) raises concerns for approximately 23% of the global population, primarily due to religious beliefs [9,10]. Therefore, other hydrocolloids, including gums, agar-agar, carrageenan, or gelatin from alternative sources like fish, are used in yogurt manufacture. However, the alternative hydrocolloids are not on par with gelatin for yogurt attributes [5,9]. In addition, the short supply and high cost of many gum-based hydrocolloids have been a major concern [11].

The recent development of a novel gel by Chelikani and Mohan combines two GRAS (Generally Recognized as Safe) status substances and demonstrates good rheological and textural properties on incorporation in yogurt [10,12]. One of the components forming the gel is disodium 5-guanylate (DG), commonly used as a flavor enhancer and approved as a food additive by the FDA (E627). DG has been used to replace or reduce sodium content in dairy and food products such as processed American cheese [13], cheddar cheese [14], and fermented sausages [15]. A recent meta-analysis has highlighted the applications of umami tasting compounds such as monosodium glutamate, disodium guanylate, disodium inosinate, and calcium diglutamate [16] for sodium reduction in food. Therefore, DG has been majorly utilized in sodium-reduced products since it provides umami taste and can mask the saltiness [16]. From our previous findings, DG forms gels in combination with organic acids, including lactic acid (LA). In yogurt, lactic acid is produced by lactic acid bacteria (LAB) and can form a gel by slowly interacting with DG in the yogurt system. The gelling mechanism is the formation of hydrogen bonds between guanylate (N–H) and lactic acid (C=O), mainly involving the OH substituents from the acid [10]. In the previous study, the formation of hydrogen bonds between DG and LA has been confirmed with FTIR. Similarly, proton NMR analysis confirmed the intermolecular hydrogen bonding in the novel gel between DG and organic acid [10,12]. In terms of the molecular structure of DG, it can be a hydrogen bond donor or acceptor; specifically, the hydrogen bond donor and acceptor count for DG is 11 and 5, respectively [17]. During the interaction of DG and organic acid, in our previous work, we found that two molecules of citric acid can form hydrogen bonds with one molecule of DG [10,12]. Thus, our objective is to compare the textural, rheological, and sensory properties of yogurt with novel gel and control yogurts prepared without using hydrocolloid, gelatin, and agar-agar as separate treatments.

## 2. Materials and Methods

### 2.1. Preparation of Novel Gel

Gel was prepared using disodium guanylate (DG) (Sigma-Aldrich, St Louis, MO, USA# 5550-12-9) and lactic acid (LA) (Sigma-Aldrich # 50-21-5) dissolved in RO water (5.34% *w*/*v* DG and 1.17% *w*/*v* of LA in solution), based on a previous study and preliminary work [10]. The suspension was mixed using a vortex mixer for 15 s, followed by heating to 50 °C in a water bath to melt and homogenize the formed gel, before cooling the prepared gel to room temperature (22 °C) for 1 h for use in the preparation of yogurt.

### 2.2. Preparation of Yogurt

Pasteurized whole milk was procured from the supermarket (general composition with protein 3.1%, fat 3.3%, and lactose 4.7%). The comprehensive yogurt manufacturing process is illustrated in Figure 1, and the treatment (with and without gelling agents) description of the different yogurt samples is included in Table 1. Specifically, the yogurt samples were control without gelling agent (Y), with agar-agar (YA), with gelatin (YG), with added lactic acid (added lactic acid concentration similar to YXL yogurt; referred to as YL) and with disodium guanylate (referred to as YX for yogurt with DG addition rate of 0.83%,), and with novel gel (referred to as YXL for yogurt with novel gel formed with DG on combination with lactic acid for an addition rate of 1% in the yogurt). The rate of addition of gelatin and agar-agar was decided based on the common rates of addition [8,18]. The gelling ability of DG in combination with lactic acid was ascertained from trials, and the rate of addition was identified as 0.83% based on good gelling properties. The rate of addition of the novel gel (combination of DG and LA) of 1% in yogurt ensures that the same amount of DG is present in YX and YXL samples. To prepare the yogurt treatment samples with the added novel gel (YXL), initially, the novel gel and the milk were separately heated to 60 °C. Subsequently, the melted gel was added to the warm milk with constant agitation, ensuring complete dispersion at 60 °C. For all yogurt samples (Table 1), the milk was heated to 60 °C, followed by the addition of the gelling agents or lactic acid for the respective yogurt samples. The specific concentration of the additives and the treatment information are detailed in Table 1. Once the dissolution of gelling agents/lactic acid was achieved by continuous agitation, milk was pasteurized by heating to 65 °C for 30 min in a hot water bath. Subsequently, it was cooled to 43 °C for the inoculation of the yogurt DVS (Direct Vat Set) starter culture, YoFlex^®^Mild 1.0 (CHR Hansen, Hoersholm, Denmark), at a rate of 50 units per 250 L of milk for all samples. Following inoculation, all the yogurt samples were transferred to plastic cups with lids and incubated at 43 °C until the pH reached around 4.5. Once this endpoint of fermentation was reached, the samples were stored in a refrigerator to arrest the fermentation process before subsequent analysis [19]. Analyses were conducted at four different time points during storage: 1, 4, 8, and 14 days. Three treatment replicates (*n* = 3) were studied for all yogurt sample analyses. Three measurements from each treatment replicate were performed for all analyses except for color (5 measurements for each treatment replicate) and texture analysis (1 measurement for each treatment replicate).

### 2.3. Acidity and pH

The acidity of yogurt samples was measured using a standard titratable acidity procedure, wherein 5 mL of yogurt was weighed in a 100 mL conical flask and 10 mL of RO water was added to mix the sample homogeneously. Five drops of 1% phenolphthalein solution (Sigma-Aldrich # 77-09-8) (1 g of phenolphthalein dissolved in 50 mL of 96% ethanol and diluted to 100 mL with RO water) were added and mixed well. The yogurt samples were then titrated against 0.1 N NaOH (Sigma-Aldrich # 1310-73-2) until a faint pink color appeared [20,21].

Titratable acidity (% LA) was calculated using$$Percentage\;of\;Titrable\;Acidity=\frac{mL\;*\;N\;*\;9}{V}$$
where N is the normality of the NaOH used, mL is the volume of 0.1 NaOH used in mL, and V is the volume of the yogurt sample used in mL.

The pH of all samples was measured using an Eutech digital pH meter (Thermo Fisher Scientific, Waltham, MA, USA), which was calibrated against buffer solutions of pH 4.0 and 7.0.

### 2.4. Syneresis

Syneresis of yogurt was determined using the centrifugation method, wherein 30 g of yogurt sample was centrifuged at 222× *g* for 10 min at 4 °C [22]. The clear supernatant was collected and weighed to calculate syneresis using the formula:$$Syneresis\;(\%)=\frac{Weight\;of\;supernatant\;(g)}{Weight\;of\;yogurt\;sample\;(g)}*100$$

### 2.5. Textural Analysis (Brittleness and Rupture Strength)

Yogurt samples (50 mL) were prepared in plastic containers and stored at 10 °C to carry out the penetration test. A penetration test was conducted to analyze the rupture strength and brittleness of samples using a texture analyser (TA-XT2, Stable Micro Systems, Godalming, UK) with a 5 kg load cell. Samples were tested for brittleness and rupture strength immediately after removing them from the refrigerator stored at 10 °C. Brittleness (mm) and rupture strength (g-force) were calculated using the Exponent software (Version 6.2). Brittleness is the distance the probe needs to penetrate before the first fracture in the gel, while the rupture strength determines the force required to deform the gel structure [23]. A cylindrical probe with a ½ diameter was used at a constant speed of 1 mm/s to penetrate a distance of 4 mm into the gel structure, once the trigger force of 5 g was attained.

### 2.6. Rheological Analysis

Apparent viscosity was analysed using rheometer Lamy RM-100, DIN 33 system (Lamy Rheology, Lyon, France) with the bob and cup type assembly (DIN 1 tube and MK DIN-2 spindle). Yogurt samples (50 mL) prepared in plastic containers and stored in refrigerated conditions were stirred in the cup for 40 s to homogenize before transferring to the DIN-1 tube to measure viscosity [24]. Apparent viscosity was measured at the shear rates 4.5, 9.1, 15.2, 38, 60.8, 76, 91.2, 121, 136, 167, and 197 s^−1^ with 60 s at each shear rate. Flow curves were plotted for all samples, and the data were fit with a power law model (Ƭ = Kγn, where Ƭ is the shear stress, k is the consistency coefficient, γ is the shear rate, and N is the flow behavior index) using the MS Excel solver add-in.

### 2.7. Microstructure Imaging

Micrographs of yogurt samples were imaged using confocal laser scanning microscopy (CLSM) (Zeiss LSM 510 Meta, Jena, Germany). Protein-specific stain Fluorescein Isothiocyanate (FITC) (Sigma-Aldrich # 27072-45-3) was prepared in ethanol (Sigma-Aldrich # 64-17-5) at a 1 mg/mL concentration. About 20 μL of dye solution was mixed with 50 μL of yogurt sample and kept for 30 min, followed by applying the yogurt–dye mixture to the microscopic slide and covering the slide using a cover slip. The excitation and emission wavelengths were 488 nm and 495 to 559 nm, respectively [25].

### 2.8. Color Values

Color values were measured using a Minolta CR-210 Chroma Meter (Minolta Camera Co. Ltd., Osaka, Japan). Yogurt samples in the plastic cups were placed horizontally on a flat surface, and the measurements were taken. Specifically, the L*, a*, and b* values are employed as indicators of color, where L* represents the lightness of the color, ranging from 0 (complete darkness) to 100 (complete lightness); a* indicates the redness or greenness of the color, with positive values (+60) representing red and negative values (−60) representing green; and b* indicates the yellowness or blueness of the color, with positive values (+60) representing yellow and negative values (−60) representing blue [26].

### 2.9. Sensory Analysis

A basic sensory analysis was conducted to compare the sensory properties of the yogurt with the novel gelling agent to the control yogurt without added gelling agents (Y) on day 1 of storage. Both samples were added with natural lemon flavor (Hansell’s lemon essence, New World, Lincoln, New Zealand) to evaluate the possibility of creating a commercially acceptable yogurt with the novel gelling agent, while masking the umami flavor of DG in YXL. A total of 49 non-expert panelists participated (from 21 to 40 years of age, consisting of 32 females and 17 male panelists), out of which 29 panelists evaluated sample Y, and 20 panelists evaluated YXL, to avoid comparison and bias between samples by the same participant. Tests were conducted on a 9-point hedonic scale where 1, 5, and 9 indicate dislike extremely, neither like nor dislike, and like extremely, respectively. The parameters evaluated were appearance (smoothness and overall consistency), flavor (at first bite and aftertaste), and overall texture. Further, the participants were also asked whether they would purchase the product if it were available in the market.

All participants were provided the written informed consent before performing sensory analysis, and the research protocol was approved by the ethics committee at Lincoln University, NZ.

### 2.10. Statistical Analysis

Statistical analysis was performed using analysis of variance (ANOVA) with the general linear model in Minitab software (Version 20, MiniTab LLC., State College, PA, USA). The Tukey comparison test was used to compare the samples with 95% confidence limits. Sensory analysis data (9-point hedonic scale) was analyzed using the Mann–Whitney/two-tailed test at a 95% confidence limit.

## 3. Results and Discussion

### 3.1. Acidity and pH

Among all samples, no significant differences were observed in acidity during storage (*p* > 0.05). The overall mean and standard error of mean for acidity (% Lactic acid) was 1.01 ± 0.01 (Appendix A). The variation in the initial pH of all the yogurt samples could be owing to the slight variations in the break point of the fermentation. Therefore, different samples (Y, YA, YG, YL, YX, and YXL) on the same day of storage were not compared statistically. Although changes in pH during storage were compared for each sample and discussed hereafter (Table 2). From previous studies, YX and YXL have been associated with DG utilizing lactic acid to produce gel continuously throughout storage [10]. Y, YL, and YX yogurts reduced significantly in their pH values over 14 days of storage, although the change in pH of YX yogurts was lower than that of Y and YL samples over the storage period. YA and YXL indicated no statistically significant decrease in pH over the storage period; however, a similar 0.06 pH change was observed for YX over the storage period. There was a reduction in pH over the storage period for YG, albeit not significant. The utilization of lactic acid by DG associated with the gel formation in the YX yogurt creates possibilities for tailoring the rheological and textural properties, as investigated in this study.

### 3.2. Syneresis

Syneresis, characterized as the appearance of serum on the surface of yogurt, is caused by the contraction of the casein network within the gel structure [4,27]. Figure 2 presents the findings for changes in syneresis during storage across all samples. It was found that syneresis in YX (18.3%) and YXL (23.3%) was significantly lower than Y (33.7%), YA (37.5%), and YL (38.4%) from day 1 through day 14 (*p* < 0.05). Hence, the yogurt with novel gelling agent (YX and YXL) had better serum holding properties than all the other yogurts with and without gelling agents, except in comparison with the yogurt with gelatin (YG), with very low to no syneresis [6,8,27,28]. Unique syneresis properties were observed for YX and YXL, with a significant decrease in the syneresis of the yogurts on Day 14, which was not observed for the other yogurts. This phenomenon can be explained by the fine gel network formation/rearrangement in the YX and YXL yogurts with the incorporation of gelling agent DG and the novel gel (DG + LA) with uniform distribution of serum phases (as also observed from the CSLM micrographs). Free lactic acid in yogurt gels induces dissolution of calcium and inorganic phosphate from casein micelles, weakening the stability of the gel and promoting the release of serum from the gel matrix [24]. The decreasing trends for syneresis during storage in YX and YXL also suggest the utilization of the available lactic acid in the system by DG, producing a uniform gel that traps more free serum available in the gel matrix. This interaction with lactic acid is more exhaustively performed by DG in YX, compared to when additional lactic acid is added in YXL, indicating the lower syneresis values of YX compared to YXL throughout storage (*p* < 0.05). This phenomenon is also evident in YL with the highest acidity, exhibiting the highest syneresis among all the yogurts (*p* < 0.05). However, among all yogurts, YG exhibited the lowest syneresis (*p* < 0.05), showcasing the highest water binding and enhanced gel viscosity of gelatin throughout storage. Previously, various studies utilizing several conventional and novel hydrocolloids have reported an increase in water holding capacity and reduced syneresis during storage owing to the water-binding ability of hydrocolloids such as xanthan gum [29], hyaluronic acid [30,31], and γ-polyglutamic acid [32]. For instance, the addition of hyaluronic acid (0.9% *w*/*v*) reduced syneresis by 27.5% and 19.28% [30], while the novel gelling agent (YXL) in our study reduced syneresis to 30% and 49% from the control on day 1 and day 14 of storage, respectively. Additionally, γ-polyglutamic acid (0.15% *w*/*w*) [32] and xanthan gum (1% *w*/*w*) [29] reduced syneresis by 46.2% and 56% on day 1 from the control, comparable to novel gelling agent samples (YX and YXL). In summary, the novel gel and DG had high water binding ability in yogurt, as indicated by lower syneresis as compared to the control and gelling agents like agar-agar throughout storage. This highlights the unique capabilities of the novel gelling agent as a potential non-animal origin hydrocolloid. As expected, gelatin showed the lowest syneresis properties amongst all the gelling agents studied.

### 3.3. Brittleness and Rupture Strength

The textural properties, such as rupture strength and brittleness, play a crucial role in determining its quality and overall acceptability. The arrangement of proteins within the gel matrix is the primary factor that defines the texture of yogurt [4]. Table 3 shows the changes in brittleness and rupture strength of the yogurt samples during storage. YX, YXL, and YG have significantly higher brittleness values as compared to control yogurt Y and other yogurts on day 1 (*p* < 0.05). A higher brittleness depicts the stability and strength of the protein network in yogurt. The mechanism of gel formation by DG with LA is primarily based on the formation of hydrogen bonds between guanylate and lactic acid. The formation of hydrogen bonds allows for water trapping, leading to reduced syneresis, higher brittleness, and rupture strength [9,10], enabling the novel gelling agent to form stronger networks in YX and YXL yogurts compared to the other samples (Y, YA, and YL). Similarly, YG is stabilized by strong electrostatic interactions with milk proteins, enabling network formation, which can be associated with its brittleness, similar to YX and YXL. Furthermore, YX (65.34 and 81.3 g-force on day 1 and 14) and YXL (76.16 and 91.17 g-force on day 1 and 14) had 6 to 7 times higher rupture strength compared to all the other yogurts studies including YG (13.81 and 23.81 g-force on day 1 and 14) (*p* < 0.05, Table 3). The increase in rupture strength for YX, YXL, and YG during storage indicates the formation of more interactions and finer networks with storage in these yogurt samples. In conclusion, lower syneresis, higher brittleness, and rupture strength during storage suggest that DG forms strong interactions with milk proteins that enable the formation of strong and fine networks with proteins during storage, which can resist deformation. These unique capabilities of novel gel yogurt could be utilized for the retention of the gel structure during transportation and storage.

### 3.4. Rheological Analysis

Yogurt displays a non-Newtonian shear-thinning behavior, characterized by a decrease in viscosity with increasing shear rate [6]. This behavior is demonstrated in Figure 3, where all samples show a decrease in viscosity with an increasing shear rate. Viscosity values at 4.5 s^−1^ and 60.8 s^−1^ are presented in Table 4, with 60.8 s^−1^ chosen as the closest approximation to the oral shear rate of 50 s^−1^ [8], and 4.5 s^−1^ representing the manual scooping shear rate. YXL and YX have comparable viscosities to Y, YA, and YL at 4.5 s^−1^ shear rate, and two times lower viscosity compared to YG at both 4.5 s^−1^ and 60.8 s^−1^ (*p* < 0.05) throughout storage. Although YX and YXL have higher rupture strength and brittleness, their viscosity rapidly decreases with increasing shear rates once the gel has been ruptured or fractured (Figure 3), increasing its ease to flow [33]. The lower viscosity for YX and YXL at higher shear rates (Table 4) also indicates improved sensory attributes such as oral texture and mouthfeel, which can be correlated with our sensory texture and smoothness findings, discussed in later sections (Section 3.7). YG consistently demonstrated the highest apparent viscosity across all storage days (Figure 3) and shear rates, which indicates the formation of a strong 3-D network and interactions with milk proteins by gelatin, that resists flow [6,8].

The power law model was used to further analyze the flow properties, which comprise the consistency coefficient (k) and flow behavior index (n) values (Table 4). These values (k and n) are crucial in evaluating product acceptability and deviation from Newtonian flow behavior, respectively [33]. YX and YXL had similar consistency coefficients as Y and YG (*p* < 0.05), as indicated by the power law model, correlating with the average viscosity behavior. All samples exhibited shear thinning behavior (*n* < 1), indicating a decrease in viscosity with an increasing shear rate. Notably, both YXL and YX showed significantly lower n values (*p* < 0.05) as compared to Y, YG, YA, and YL throughout storage. YXL, in particular, exhibited the lowest n values (0.13 to 0.16), signifying distinct flow behavior requiring only a slight increase in shear rates to effect rapid reduction in the resistance to flow, compared to other samples [8,9]. Similarly, recent studies utilizing hyaluronic acid [30], γ-polyglutamic acid [32], and micro-nano fibrillated cellulose [34] in yogurt led to comparable or higher viscosities than the control, owing to increased water binding and formation of a strong gel network with the use of these novel hydrocolloids. For instance, the enrichment of yogurt with hyaluronic acid (0.8% *w*/*v*) reduced the n value from 0.68 to 0.49 and increased the k value from 2.14 to 2.3 [30]. This is a similar trend observed in our study, with the addition of novel gel (YX and YXL), yogurts were more shear-thinning with enhanced consistency index. In another study, an increase in concentration of γ-polyglutamic acid from 0 to 0.3% (*w*/*w*) enhanced the shear thinning behavior of yogurt, with the n value of yogurt increasing from 0.162 in the control to 0.135, respectively [32], comparable to our results.

Overall, the addition of novel gel to yogurt increased its pseudoplastic or shear-thinning behavior, compared to other yogurt samples, with and without other gelling agents, throughout storage. Higher pseudoplasticity (lower n values) is associated with lower shearing energy needed for enhanced mixability and pumpability. The texture and rheological studies indicate that the gelling activity of DG in yogurts leads to the formation of a strong and brittle gel network that requires high yield stress or rupture strength for breaking the network structure. However, once this network is disrupted by mixing and agitation, these yogurts can flow easily and can be pumped energy efficiently, making the novel gelling agent suitable for application in both set and stirred yogurts. Additionally, studies are needed about the reforming of the gel network and the strength of the reformed gels for the DG-based novel gelling agent in yogurts.

### 3.5. Microstructural Analysis

The microstructural images show the protein regions in green and void spaces in black, representing the serum portion (Figure 4). A comparison of microstructures was made for all yogurts on day 1 and day 14 of storage. On days 1 and 14, YXL and YX exhibited a more uniform protein network with a well-distributed serum phase compared to all the other yogurts. This can be related to the lower syneresis (Figure 2) and higher rupture strength (Table 3) of these samples compared to Y, YA, and YL. Therefore, utilization of lactic acid produced by the lactic acid bacteria in yogurts by DG, throughout storage in YX and YXL, enables the formation of hydrogen bonds and trapping of water (less syneresis), leading to a strong and uniform protein network.

### 3.6. Color Values

The color of the final product plays a crucial role in consumer acceptability. The results for color parameters are detailed in Table 5. On day 1, YX, YXL, YG, and YL showed significantly higher lightness (L*) values compared to the control samples Y and YA (*p* < 0.05). The results for L* value can be correlated with the uniform network and the lower serum pockets/separation in the microstructure (CLSM images, see Figure 4) that scatter more light, for YX, YXL, and YG as compared to Y and YA [35]. Similarly, the significantly lower values of yellowness (b*) for YX and YXL can be associated with the slightly yellow color of the serum phase existing as smaller pockets in these samples compared to the others. In terms of redness (a*), no significant differences were observed in redness values among Y, YA, YG, YL, YX, and YXL during storage, except for YL, which had a lower value on day 1 (*p* < 0.05). Overall, yogurts containing novel gel (YX and YXL) had acceptable color, producing a whiter appearance compared to other samples in the study.

### 3.7. Sensory Analysis

In the sensory analysis depicted in Figure 5, YXL did not exhibit any significant difference from the control yogurt Y (*p* > 0.05) when both had lemon flavor. The flavor added could likely mask the umami flavor associated with DG, which is associated with its general use as a flavor enhancer. YXL received higher average ratings in terms of appearance (smoothness: 5.8 vs. 5.24, overall consistency: 5.1 vs. 4.82) and overall texture (5.1 vs. 4.8), albeit not significantly. On the other hand, YXL rated lower in terms of flavor (taste at first bite: 3.1 vs. 4.06 and aftertaste: 3.85 vs. 4.51) compared to the control (Y). The observations indicate that YXL produced yogurt with acceptable appearance and texture and was comparable to control yogurt (without gel addition; Y), which correlates well with higher L values and viscosity and texture values of YXL. However, the taste and aftertaste were rated lower in the YXL sample. Moreover, in terms of willingness to purchase the product by the participants, 20% of participants found YXL acceptable, while only 17% of the participants found Y acceptable. Our findings provide a preliminary indicator of the comparability of the yogurt with a novel gelling agent to the control yogurt and its relative consumer acceptability. More research is required to develop suitable natural flavors and flavoring compounds that could enhance and/or mask the umami-associated flavor of the novel gelling agent.

## 4. Conclusions

This study aimed to evaluate the feasibility of a novel gel, composed of DG and LA, as a prospective substitute for commonly used hydrocolloids, specifically gelatin, in yogurt production. The yogurt formulated with the novel gel exhibited superior mechanical properties, with the highest rupture strength and brittleness among all samples. Notably, it displayed reduced syneresis and sensory attributes comparable to the control. Microstructural analysis indicated similarities with yogurt prepared using other hydrocolloids, suggesting analogous gelling capabilities. Additionally, the lower viscosity at high shear rate of yogurt containing a novel gelling agent might be useful in enhanced pumpability for drinkable or high protein yogurts. Moreover, future studies are required to understand the interactions between DG and milk proteins to elucidate the novel gel role as a hydrocolloid. Further flavor chemistry research is required to optimize the taste of the yogurts manufactured with DG as a novel gelling agent, with the addition of flavoring/flavor masking compounds.

## Figures and Tables

**Figure 1 foods-14-02252-f001:**
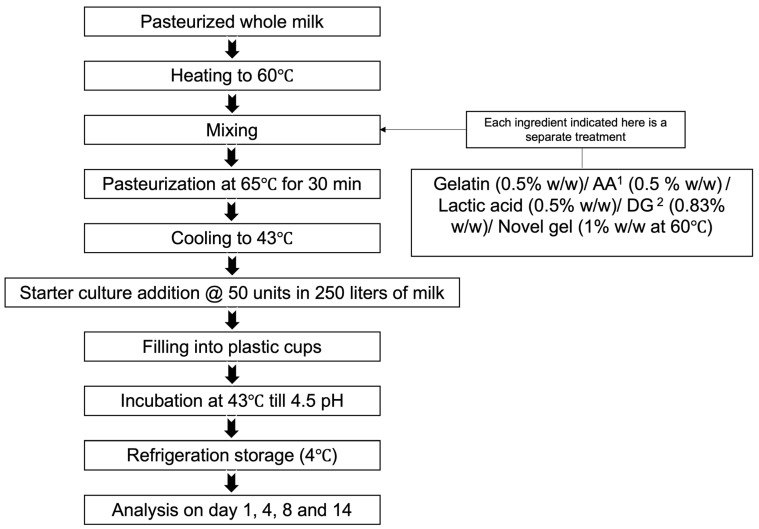
Process flow diagram for preparing all yogurt samples in the study. ^1^: AA represents agar-agar. ^2^: DG represents Disodium 5-guanylate.

**Figure 2 foods-14-02252-f002:**
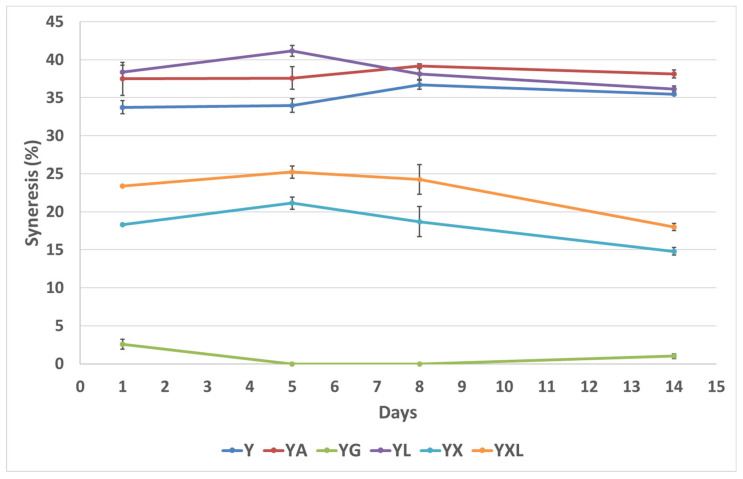
Changes in syneresis (%) during storage for all yogurt samples ^1^. ^1^: Y—control, YA—yogurt with agar-agar, YG—yogurt with gelatin, YL—yogurt with lactic acid, YX—yogurt with disodium 5-guanylate, YXL—yogurt with novel gel.

**Figure 3 foods-14-02252-f003:**
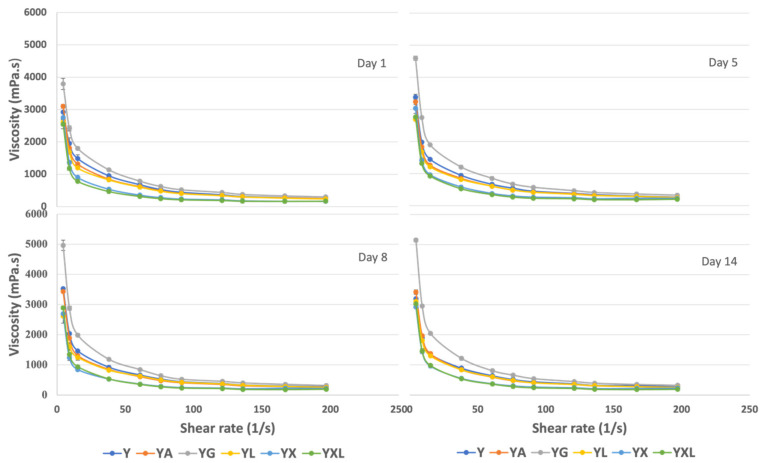
Flow curve depicting viscosity as a function of shear rate during storage for all yogurt samples ^1^. ^1^: Y—control, YA—yogurt with agar-agar, YG—yogurt with gelatin, YL—yogurt with lactic acid, YX—yogurt with disodium 5-guanylate, YXL—yogurt with novel gel.

**Figure 4 foods-14-02252-f004:**
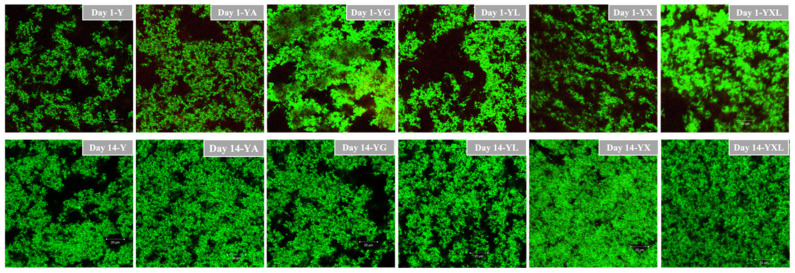
Confocal laser scanning microscopy images of all yogurt samples ^1^ compared on day 1 and day 14 of storage. ^1^: Y—control, YA—yogurt with agar-agar, YG—yogurt with gelatin, YL—yogurt with lactic acid, YX—yogurt with disodium 5-guanylate, YXL—yogurt with novel gel. The measurement bar is 20 um.

**Figure 5 foods-14-02252-f005:**
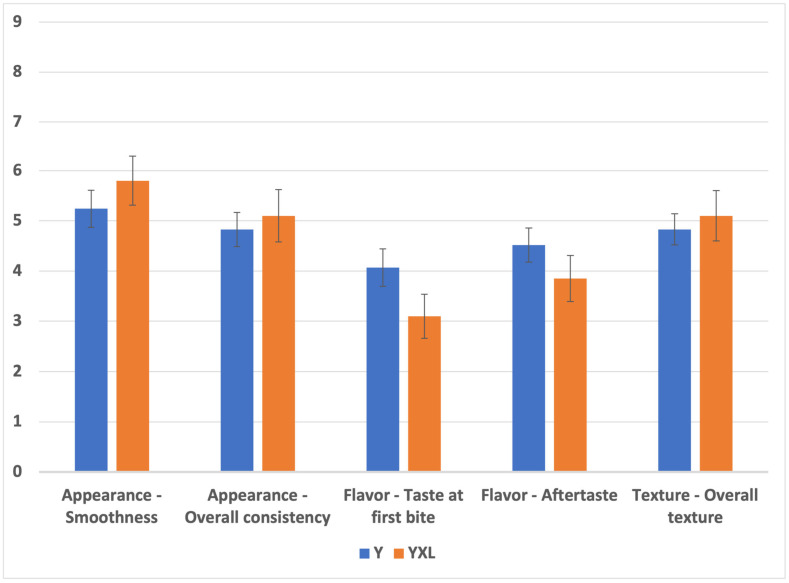
Sensory analysis comparison of control yogurt without gelling agent (Y) and treatment yogurt with novel gel (YXL). Y—control and YXL—yogurt with novel gel. Tests were conducted on a 9-point hedonic scale where 1, 5, and 9 indicate dislike extremely, neither like nor dislike, and like extremely, respectively.

**Table 1 foods-14-02252-t001:** Experimental design and treatments evaluated in the study.

Samples	Description	Percentage (*w*/*v*) Addition of Gelling Agent in Final Product	Form of Addition of Ingredient in Milk
Y	Control	-	-
YA	AA ^1^	0.5	Powder
YG	Gelatin	0.5	Powder
YL	LA ^2^	0.012	Powder
YX	DG ^3^	0.83	Powder
YXL	Novel gel	1	Gel made with LA ^1^, DG ^2^, and water

^1^ Agar-agar. ^2^ Lactic acid. ^3^ Disodium 5-guanylate.

**Table 2 foods-14-02252-t002:** Changes in pH for all samples ^1^ during storage period.

Parameter	Day	Y	YA	YG	YL	YX	YXL
pH	1	4.46 ± 0.02 ^A^	4.41 ± 0.00 ^A^	4.42 ± 0.01 ^A^	4.44 ± 0.00 ^A^	4.55 ± 0.01 ^AB^	4.48 ± 0.02 ^A^
	5	4.41 ± 0.03 ^A^	4.37 ± 0.03 ^A^	4.33 ± 0.02 ^B^	4.37 ± 0.00 ^AB^	4.59 ± 0.01 ^A^	4.46 ± 0.01 ^AB^
	8	4.31 ± 0.00 ^B^	4.33 ± 0.00 ^A^	4.38 ± 0.01 ^AB^	4.31 ± 0.01 ^B^	4.47 ± 0.00 ^B^	4.38 ± 0.00 ^B^
	14	4.27 ± 0.02 ^B^	4.34 ± 0.01 ^A^	4.41 ± 0.01 ^AB^	4.33 ± 0.03 ^B^	4.49 ± 0.01 ^B^	4.43 ± 0.02 ^AB^

Values are represented as mean ± standard error of mean. Means with different uppercase letters within the same column for the same parameter are significantly different (*p* < 0.05). Means within the same row (among samples) were not compared statistically as the break point of fermentation for the yogurt samples was slightly different, as indicated by the Day 1 data, and these variations may also persist over the storage period. ^1^: Y—control, YA—yogurt with agar-agar, YG—yogurt with gelatin, YL—yogurt with lactic acid, YX—yogurt with disodium 5-guanylate, YXL—yogurt with novel gel.

**Table 3 foods-14-02252-t003:** Changes in textural attributes (brittleness and rupture strength) for all samples ^1^ during storage period.

Parameter	Day	Y	YA	YG	YL	YX	YXL
Brittleness (mm)	1	6.35 ± 0.47 ^bA^	5.24 ± 0.23 ^cB^	7.94 ± 0.02 ^aA^	5.57 ± 0.12 ^bcB^	7.98 ± 0.01 ^aA^	7.99 ± 0.01 ^aA^
	5	4.13 ± 0.22 ^cB^	4.08 ± 0.11 ^cB^	6.83 ± 0.45 ^bB^	4.19 ± 0.17 ^cC^	7.99 ± 0.05 ^aA^	7.99 ± 0.03 ^aA^
	8	4.32 ± 0.30 ^bB^	4.24 ± 0.34 ^bB^	7.65 ± 0.14 ^aAB^	7.73 ± 0.19 ^aA^	7.95 ± 0.04 ^aA^	7.98 ± 0.02 ^aA^
	14	3.94 ± 0.23 ^bB^	8 ± 0.00 ^aA^	7.29 ± 0.02 ^aAB^	7.94 ± 0.02 ^aA^	7.97 ± 0.01 ^aA^	7.99 ± 0.03 ^aA^
Rupture strength (g-force)	1	12.13 ± 0.75 ^cA^	15.58 ± 0.13 ^cA^	13.81 ± 0.85 ^cB^	12.47 ± 0.20 ^cA^	65.34 ± 3.92 ^bB^	76.16 ± 1.11 ^aB^
	5	17.82 ± 0.31 ^cA^	20.17 ± 0.19 ^cA^	23.71 ± 0.31 ^cdA^	15.34 ± 0.58 ^dA^	75.73 ± 1.05 ^bA^	94.04 ± 2.43 ^aA^
	8	16.6 ± 0.19 ^cA^	19.29 ± 0.85 ^cA^	24.07 ± 0.26 ^cA^	16.11 ± 0.79 ^dA^	76.48 ± 1.97 ^bA^	91.75 ± 0.59 ^aA^
	14	15.43 ± 0.63 ^cA^	19.46 ± 0.51 ^cA^	23.81 ± 0.57 ^cA^	15.46 ± 0.57 ^dA^	81.3 ± 2.75 ^bA^	91.17 ± 1.64 ^aA^

Values are represented as mean ± standard error of mean. Means with different lowercase letters within the same row are significantly different (*p* < 0.05). Means with different uppercase letters within the same column for the same parameter are significantly different (*p* < 0.05). ^1^: Y—control, YA—yogurt with agar-agar, YG—yogurt with gelatin, YL—yogurt with lactic acid, YX—yogurt with disodium 5-guanylate, YXL—yogurt with novel gel.

**Table 4 foods-14-02252-t004:** Changes in apparent viscosity (Pa.s) at two different shear rates (4.5 and 61 s^−1^) ^1^ and rheological parameters (k and n) obtained from flow curves by fitting the power law model for all samples ^2^ during storage.

Shear Rate/ Parameters	Day	Y	YA	YG	YL	YX	YXL
4.5 s^−1^	1	2.92 ± 0.21 ^bA^	3.09 ± 0.05 ^bA^	3.79 ± 0.17 ^aB^	2.61 ± 0.12 ^bA^	2.75 ± 0.15 ^bA^	2.54 ± 0.14 ^bA^
	5	3.38 ± 0.08 ^bA^	3.24 ± 0.08 ^bcA^	4.59 ± 0.06 ^aA^	2.68 ± 0.01 ^cA^	3.03 ± 0.15 ^bcA^	2.76 ± 0.09 ^cA^
	8	3.52 ± 0.03 ^bA^	3.43 ± 0.02 ^bcA^	4.96 ± 0.17 ^aA^	2.61 ± 0.23 ^cA^	2.69 ± 0.02 ^cA^	2.88 ± 0.06 ^cA^
	14	3.19 ± 0.05 ^bA^	3.41 ± 0.07 ^bA^	5.14 ± 0.06 ^aA^	3.10 ± 0.03 ^bA^	2.92 ± 0.05 ^bA^	3.02 ± 0.09 ^bA^
60.8 s^−1^	1	0.67 ± 0.01 ^bA^	0.61 ± 0.09 ^cA^	0.78 ± 0.01 ^aB^	0.60 ± 0.01 ^cA^	0.35 ± 0.03 ^dA^	0.31 ± 0.04 ^dA^
	5	0.66 ± 0.03 ^bA^	0.60 ± 0.09 ^cA^	0.84 ± 0.06 ^aA^	0.60 ± 0.01 ^bcA^	0.37 ± 0.05 ^dA^	0.34 ± 0.01 ^dA^
	8	0.66 ± 0.03 ^bA^	0.60 ± 0.01 ^bA^	0.84 ± 0.06 ^aA^	0.63 ± 0.01 ^bA^	0.36 ± 0.08 ^cA^	0.35 ± 0.01 ^cA^
	14	0.64 ± 0.05 ^bA^	0.59 ± 0.05 ^bA^	0.80 ± 0.03 ^aAB^	0.60 ± 0.08 ^bA^	0.37 ± 0.05 ^cA^	0.36 ± 0.05 ^cA^
k ^3^	1	7.25 ± 0.76 ^abA^	8.18 ± 0.17 ^abA^	9.85 ± 0.61 ^aB^	6.42 ± 0.45 ^bA^	9.43 ± 0.80 ^abAB^	9.20 ± 1.01 ^abA^
	5	8.91 ± 0.37 ^bcA^	8.82 ± 0.42 ^bcA^	12.73 ± 0.28 ^aAB^	6.61 ± 0.03 ^cA^	10.48 ± 0.87 ^abA^	9.48 ± 0.05 ^bcA^
	8	9.56 ± 0.19 ^bA^	9.67 ± 0.08 ^bA^	14.52 ± 0.73 ^aA^	6.23 ± 0.79 ^cA^	6.96 ± 1.64 ^bcA^	10.15 ± 0.13 ^bA^
	14	8.32 ± 0.25 ^bA^	9.59 ± 0.30 ^bA^	15.38 ± 0.27 ^aA^	8.25 ± 0.12 ^bA^	9.81 ± 0.21 ^bAB^	10.77 ± 0.43 ^bA^
n ^3^	1	0.40 ± 0.01 ^aA^	0.34 ± 0.03 ^aA^	0.36 ± 0.01 ^aA^	0.39 ± 0.01 ^aA^	0.16 ± 0.01 ^bB^	0.13 ± 0.03 ^bA^
	5	0.34 ± 0.01 ^aA^	0.31 ± 0.01 ^aA^	0.31 ± 0.06 ^aA^	0.39 ± 0.04 ^aA^	0.15 ± 0.01 ^bB^	0.16 ± 0.04 ^bA^
	8	0.32 ± 0.06 ^abA^	0.29 ± 0.07 ^bA^	0.28 ± 0.01 ^bA^	0.41 ± 0.02 ^aA^	0.27 ± 0.07 ^bA^	0.14 ± 0.05 ^cA^
	14	0.35 ± 0.09 ^aA^	0.30 ± 0.04 ^aA^	0.26 ± 0.05 ^bA^	0.33 ± 0.04 ^aA^	0.17 ± 0.03 ^bcAB^	0.13 ± 0.07 ^cA^

Values are represented as mean ± standard error of mean. Means with different lowercase letters within the same row are significantly different (*p* < 0.05). Means with different uppercase letters within the same column for the same parameter are significantly different (*p* < 0.05). ^1^: 4.5 s^−1^ and 60.8 s^−1^ represent manual scooping shear rate and oral mixing shear rate, respectively. ^2^: Y—control, YA—yogurt with agar-agar, YG—yogurt with gelatin, YL—yogurt with lactic acid, YX—yogurt with disodium 5-guanylate, YXL—yogurt with novel gel. ^3^: k and n represent consistency coefficient and flow behavior index, respectively, calculated using the power law model with an R^2^ value of 0.99 for all samples.

**Table 5 foods-14-02252-t005:** Changes in color values for all samples ^1^ during storage.

Parameter ^2^	Day	Y	YA	YG	YL	YX	YXL
L*	1	28.3 ± 1.2 ^cA^	33.1 ± 0.6 ^bA^	37.1 ± 0.7 ^aA^	40.7 ± 1.2 ^aA^	37.7 ± 0.6 ^aA^	37.8 ± 1.2 ^aA^
	5	36.2 ± 0.4 ^aB^	37.4 ± 1.4 ^aB^	37.0 ± 4.1 ^aA^	39.1 ± 1.4 ^aA^	32.5 ± 1.2 ^bB^	37.7 ± 0.7 ^aA^
	8	40.5 ± 1.1 ^aB^	38.9 ± 1.4 ^abB^	40.7 ± 0.4 ^aA^	35.4 ± 0.7 ^bB^	38.2 ± 0.3 ^abA^	40.4 ± 0.43 ^aA^
	14	37.2 ± 0.6 ^abB^	33.4 ± 0.7 ^bA^	35.8 ± 0.5 ^abB^	34.0 ± 1.7 ^bB^	32.8 ± 1.5 ^bB^	38.9 ± 1.64 ^aA^
a*	1	−1.7 ± 0.1 ^abA^	−1.5 ± 0.0 ^aA^	−1.7 ± 0.1 ^abA^	−1.8 ± 0.1 ^bA^	−1.5 ± 0.1 ^aA^	−1.5 ± 0.0 ^aA^
	5	−1.6 ± 0.0 ^abA^	−1.8 ± 0.1 ^bB^	−1.6 ± 0.1 ^abA^	−1.7 ± 0.1 ^abA^	−1.5 ± 0.1 ^aA^	−1.5 ± 0.0 ^aA^
	8	−1.8 ± 0.0 ^aA^	−1.9 ± 0.1 ^aB^	−1.7 ± 0.0 ^aA^	−1.8 ± 0.0 ^aA^	−1.7 ± 0.0 ^aA^	−1.7 ± 0.0 ^aB^
	14	−1.7 ± 0.0 ^aA^	−1.6 ± 0.0 ^aA^	−1.5 ± 0.0 ^aA^	−1.6 ± 0.1 ^aA^	−1.5 ± 0.1 ^aA^	−1.6 ± 0.0 ^aA^
b*	1	7.6 ± 0.4 ^aA^	6.5 ± 0.1 ^bcA^	6.8 ± 0.2 ^bA^	7.4 ± 0.1 ^aA^	6.4 ± 0.2 ^bcA^	6.1 ± 0.1 ^cA^
	5	6.8 ± 0.0 ^aB^	7.0 ± 0.2 ^aB^	6.8 ± 0.4 ^aA^	7.1 ± 0.1 ^aA^	6.5 ± 0.2 ^abA^	6.3 ± 0.1 ^bA^
	8	7.2 ± 0.1 ^aAB^	7.2 ± 0.2 ^aB^	7.2 ± 0.0 ^aA^	7.0 ± 0.1 ^abA^	6.6 ± 0.1 ^abA^	6.4 ± 0.04 ^bA^
	14	6.9 ± 0.1 ^aB^	6.6 ± 0.1 ^aB^	6.9 ± 0.1 ^aA^	6.9 ± 0.3 ^aA^	6.2 ± 0.4 ^bA^	6.1 ± 0.1 ^bA^

Values are represented as mean ± standard error of mean. Means with different lowercase letters within the same row are significantly different (*p* < 0.05). Means with different uppercase letters within the same column for the same parameter are significantly different (*p* < 0.05). ^1^: Y—control, YA—yogurt with agar-agar, YG—yogurt with gelatin, YL—yogurt with lactic acid, YX—yogurt with disodium 5-guanylate, YXL—yogurt with novel gel. ^2^: L, a, and b represent lightness, redness, and yellowness of the sample, respectively.

## Data Availability

The original contributions presented in the study are included in the article/Appendix A, further inquiries can be directed to the corresponding author.

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
