# Peer review of "Evaluating a Novel Hydrocolloid Alternative for Yogurt Production: Rheological, Microstructural, and Sensory Properties"

_foods, 2025, doi:10.3390/foods14132252_

Round 1
Reviewer 1 Report
Comments and Suggestions for Authors
The manuscript explored the feasibility of using a new type of gel composed of disodium 5-guanylate (DG) and lactic acid (LA) as an alternative to traditional emulsifiers. There are some critical concerns of the manuscript which should be addressed by the authors.
- In the “3 Results and discussion” part, which I found is very superficial and need more discussion with comparison from existing literature.
- Sensory confirmation is lacking for the sensory analysis part. Please note that approval in part “9 Sensory analysis”.
- The manuscript mentions that sensory analysis was conducted by non-expert panelists, but it does not elaborate on the training situation, and the specific guidance during the evaluation process.
- For the references, the better explanation and comparison among the experiment results such as the similarities and differences were lack, which needs further improvement.
- Line 65, The gelling mechanism of DG and LA need a more detailed description.
- Line 66-70 can be deleted, which is not smooth with the context before and after.
- The paper analyzed the characteristic changes of yogurt within a 14-day storage period. However, there are relatively few studies on the impact of longer-term storage (such as several weeks or months) on the quality of yogurt. This part of the research can be carried out in the future.
- The authors mentioned that DG is a flavor enhancer (Line 61). The potential flavor changes that the new gel might introduce in yogurt can be carried out in the future.
Author Response
|
Reviewer 1 |
Response |
|
The manuscript explored the feasibility of using a new type of gel composed of disodium 5-guanylate (DG) and lactic acid (LA) as an alternative to traditional emulsifiers. There are some critical concerns of the manuscript which should be addressed by the authors. |
Thanks for the suggestions for improving the manuscript. |
|
In the “3 Results and discussion” part, which I found is very superficial and need more discussion with comparison from existing literature. |
Considering the reviewer’s suggestion, we have now compared with recent studies using novel hydrocolloids in relevant sections such as syneresis and rheological measurements. L289-297, 351-353, 389-398 |
|
Sensory confirmation is lacking for the sensory analysis part. Please note that approval in part “9 Sensory analysis” |
The documents regarding ethical approval has been provided to the editor and now we have included this information in manuscript as well based on your suggestion. Also, we have added more details and improved sensory analysis methods section. L211-227 |
|
The manuscript mentions that sensory analysis was conducted by non-expert panelists, but it does not elaborate on the training situation, and the specific guidance during the evaluation process. |
The details of the product information sheet, and sensory information ethical approval documentation has been submitted to the editor. The sensory analysis was a 9-point hedonic scale consumer liking study, which does not involve any training of the participants, apart from the information provided in the product information sheet to each participant in the sensory study. we have added more details and improved sensory analysis methods section L211-227 |
|
For the references, the better explanation and comparison among the experiment results such as the similarities and differences were lack, which needs further improvement. |
We have improved the discussion now, based on other reviewer’s comments and also supported with recent literature. L 63-82, 289-297, 351-353, 389-398 |
|
Line 65, The gelling mechanism of DG and LA need a more detailed description |
We have added more details and included the reference to the previous study wherein the spectral analysis was used to confirm the mechanism. L 63-82 |
|
Line 66-70 can be deleted, which is not smooth with the context before and after. |
We have removed Line 66-70. |
|
The paper analyzed the characteristic changes of yogurt within a 14-day storage period. However, there are relatively few studies on the impact of longer-term storage (such as several weeks or months) on the quality of yogurt. This part of the research can be carried out in the future. |
We appreciate your suggestions for future studies. |
|
The authors mentioned that DG is a flavor enhancer (Line 61). The potential flavor changes that the new gel might introduce in yogurt can be carried out in the future. |
Thanks for pointing it out as a future study. |
Reviewer 2 Report
Comments and Suggestions for Authors
- In the introduction, the authors could provide more information about disodium 5-guanylate, its use in food, and its interaction with different organic acids.
- Why did the authors use different percentages (w/w) of the gelling agent in each formulation for the experiment design in Table 1?.
- The authors reported the formation of a gel through the interaction of lactic acid produced by lactic acid bacteria (LAB) and disodium 5-guanylate. How many moles of lactic acid are required for a reaction with disodium 5-guanylate, and vice versa?.
Author Response
|
Reviewer 2 |
|
|
In the introduction, the authors could provide more information about disodium 5-guanylate, its use in food, and its interaction with different organic acids. |
Thanks for the suggestion, we have included recent research on disodium guanylate in introduction. L 63-82. |
|
Why did the authors use different percentages (w/w) of the gelling agent in each formulation for the experiment design in Table 1?. |
This was chosen based on preliminary work done using novel gel. We have included that in the manuscript (L100-110). Also, corrected Table 1 that rate of addition was w/v not w/w. With regards to concentrations of other gelling agents such as agar agar and gelatin we chose most commonly used concentration (0.5 % w/w) and included references L107. |
|
The authors reported the formation of a gel through the interaction of lactic acid produced by lactic acid bacteria (LAB) and disodium 5-guanylate. How many moles of lactic acid are required for a reaction with disodium 5-guanylate, and vice versa? |
Appreciate the comment. We have included this information in L 72-78, elaborating more on interaction. |
Reviewer 3 Report
Comments and Suggestions for Authors
1.The table format is incorrect; it should follow the "three-line table" style.
2. Please use Origin software to generate the experimental figures
3. Figure 5 (sensory analysis) lacks error bars.
4. Lines 95–99 this sentence is difficult to understand and need to be rewritten.
5.Line 104: Is the inoculation ratio of the yogurt starter culture too low? Did the authors base this on previous literature? If so, please include the reference in the revised version.
6. In Table 1, why is there a large discrepancy in the amount of gelling agent added to the final product between YL and YX compared to the other three groups? Was the dosage standard experimentally validated beforehand?
7.The two images on the right side of Figure 3 lack a y-axis label and need to be completed.
8.The significant digits in Table 5 are inconsistent and need to be revised.
9.The reference formatting needs to be corrected.
Author Response
|
Reviewer 3 |
|
|
The table format is incorrect; it should follow the "three-line table" style. |
We have formatted it correctly now. |
|
Please use Origin software to generate the experimental figures |
Thanks for the suggestion. We will consider this suggestion for the use of Origin software for future studies; however, we will refrain from making changes to the data representation, owing to the lack of clarity about the specific need to use a new software in this study. |
|
Figure 5 (sensory analysis) lacks error bars. |
We have revised the figure to bar graphs and included error bars. |
|
Lines 95–99 this sentence is difficult to understand and need to be rewritten. |
Thanks, we have now revised the statement. |
|
.Line 104: Is the inoculation ratio of the yogurt starter culture too low? Did the authors base this on previous literature? If so, please include the reference in the revised version. |
The % addition was based on the recommendation from supplier. Usually, the DVS cultures are supplied based on units per 1000 L of milk, hence the % w/v was calculated and reported in our study. |
|
In Table 1, why is there a large discrepancy in the amount of gelling agent added to the final product between YL and YX compared to the other three groups? Was the dosage standard experimentally validated beforehand? |
Yes, the amounts used were based on preliminary work. Further, the concentration of DG and LA in YX and YL, was based on individual components concentration in yogurt YXL (DG+LA+water). This has been mentioned in the manuscript now L 100-110. |
|
The two images on the right side of Figure 3 lack a y-axis label and need to be completed |
The y-axis scale was kept the same for all figures and therefore, two right side images in Fig 3 utilizes the same y-axis as left two images. This was done to make the visuals graphically more clear. |
|
The significant digits in Table 5 are inconsistent and need to be revised. |
We could not find any inconsistency with the significant digits or superscripts in Table 5. |
|
The reference formatting needs to be corrected. |
We have corrected all the mistakes in the reference section. |
Reviewer 4 Report
Comments and Suggestions for Authors
This manuscript evaluates a novel gelling agent, combining DG and LA, as a substitute for conventional hydrocolloids in yogurt production. Six yogurt samples (control, novel gel, DG, LA, gelatin, agar-agar) are compared for acidity, rheological, textural, microstructural, and sensory properties over 14 days. The novel gel yogurt exhibits reduced syneresis, comparable viscosity to the control at low shear rates, higher rupture strength, and a compact protein network similar to gelatin-containing yogurt, with no significant sensory differences from the control.
GENERAL COMMENTS
The study is well-structured, timely, and explores a promising alternative to conventional gelling agents. However, several aspects require improvement to enhance the scientific rigor and contextual relevance of the work:
- Lack of Contextualization with Similar Research: A major shortcoming is the limited comparison with existing studies on hydrocolloids or novel gelling agents in dairy applications. This weakens the discussion and undermines the broader scientific relevance of the findings.
- Insufficient and Outdated References: The manuscript includes only 21 references, with minimal representation from recent years (only one each from 2023 and 2024, none from 2025). The limited and outdated citations detract from the study’s alignment with current research trends.
- Methodological Limitations: While the experimental procedures are detailed, some methods lack citations to standard protocols, potentially affecting reproducibility. References to validated analytical techniques are necessary to support credibility.
- Sensory Analysis Gaps: The sensory evaluation includes only two yogurt samples, and no overall acceptability score. The rationale for these choices is unclear and should be addressed to improve the interpretability of sensory data.
Specific Comments
- Line 29-31 (Keywords): Include “novel gelling agent” and/or “di-sodium 5-guanylate” to reflect the core focus of the study. Consider removing “gelatin” to avoid ambiguity.
- Line 70-73 (Materials and Methods - Novel Gel Preparation): The chosen concentrations of DG (5.34% w/v) and LA (1.17% w/v) are not justified. Provide a rationale or cite reference [10] if based on prior optimization.
- Line 83 (Figure 1): Replace “heat treatment 65 °C for 30 min” with “pasteurization at 65 °C for 30 min” to reflect standard dairy processing terminology.
- Line 106-112 (Table 1): Use consistent abbreviations throughout (e.g., “AA” for agar-agar) to avoid confusion between Figure 1 and Table 1.
- Line 111 (Methods - Textural Analysis): Only one measurement per replicate was conducted for texture analysis, unlike other analyses (three measurements). Justify this choice or discuss its impact on data reliability.
- Line 124-130 (Methods - Acidity): The titratable acidity method lacks a standard reference (e.g., AOAC). Cite a validated protocol to ensure reproducibility.
- Line 149-154 (Methods - Rheological Analysis): Explain the choice of 4.5 s⁻¹ (manual scooping) and 60.8 s⁻¹ (oral mixing) with references linking to yogurt texture perception.
- Line 182-190 (Sensory Analysis): Justify why only two samples (Y, YXL) were sensory-analyzed, excluding others (e.g., YG). Specify the testing day (e.g., day 1 or day 14). Clarify the addition of lemon flavor (10 drops) to mask umami, including concentration (drops per sample volume). Clarify why not all 49 panelists evaluated both samples (29 for Y, 20 for YXL) and why overall acceptability was not assessed, as this limits sensory insights.
- Line 202 (Supplementary Table 1): Since no statistically significant differences were observed in acidity, remove the superscript letters to prevent misinterpretation.
- Line 247-249 (Discussion): The correlation between syneresis, texture, and microstructure is discussed prematurely. Consider relocating this discussion to a section following the analysis of individual results.
- Line 261-264 (Figure 2): The x-axis shows only even days (2, 4, etc.), but measurements were taken on days 1, 5, 8, and 14. Label all days (1–15) to clearly indicate measurement points. Include statistical significance annotations (e.g., letters for p < 0.05).
- Line 321-330 (Table 4): The lower viscosity of YX and YXL at 60.8 s⁻¹ (0.35–0.37 Pa.s vs. 0.64–0.67 Pa.s for Y) suggests improved oral texture, but this is not discussed (lines 371-372). Link rheological data to sensory results (lines 400-412).
- Line 474-496 (References): The reference list is insufficient and outdated. Include more recent studies (2023–2025), particularly those addressing novel hydrocolloids, yogurt texture, or DG/LA applications. Correct reference [12] (“V Chelikani, a.M.S.M.”) for formatting and completeness.
Author Response
|
Reviewer 4 |
|
|
This manuscript evaluates a novel gelling agent, combining DG and LA, as a substitute for conventional hydrocolloids in yogurt production. Six yogurt samples (control, novel gel, DG, LA, gelatin, agar-agar) are compared for acidity, rheological, textural, microstructural, and sensory properties over 14 days. The novel gel yogurt exhibits reduced syneresis, comparable viscosity to the control at low shear rates, higher rupture strength, and a compact protein network similar to gelatin-containing yogurt, with no significant sensory differences from the control. |
Thanks for reviewing the manuscript and providing valuable comments for improvement. |
|
GENERAL COMMENTS The study is well-structured, timely, and explores a promising alternative to conventional gelling agents. However, several aspects require improvement to enhance the scientific rigor and contextual relevance of the work:
|
We have addressed the comments and these issues in the manuscript. Compared with recent novel hydrocolloids-based studies, added new references, added citations in methodology part, provided more details in sensory. |
|
Specific Comments
Line 29-31 (Keywords): Include “novel gelling agent” and/or “di-sodium 5-guanylate” to reflect the core focus of the study. Consider removing “gelatin” to avoid ambiguity. |
Thanks for your suggestion. We have included di-sodium 5-guanylate in the keywords L29 and removed gelatin. |
|
Line 70-73 (Materials and Methods - Novel Gel Preparation): The chosen concentrations of DG (5.34% w/v) and LA (1.17% w/v) are not justified. Provide a rationale or cite reference [10] if based on prior optimization. |
These were chosen based on preliminary trials and previous study. We have included the information in the manuscript in L 87, 100-110. |
|
Line 83 (Figure 1): Replace “heat treatment 65 °C for 30 min” with “pasteurization at 65 °C for 30 min” to reflect standard dairy processing terminology. |
Revised as suggested. |
|
Line 106-112 (Table 1): Use consistent abbreviations throughout (e.g., “AA” for agar-agar) to avoid confusion between Figure 1 and Table 1. |
Changed as suggested. |
|
Line 111 (Methods - Textural Analysis): Only one measurement per replicate was conducted for texture analysis, unlike other analyses (three measurements). Justify this choice or discuss its impact on data reliability. |
Penetration test was conducted for texture analysis with a half inch probe and only one measurement was taken from each 50mL cup for each of the three replicates. This was to ensure that the gel is undisturbed when each replicate measurement was taken. |
|
Line 124-130 (Methods - Acidity): The titratable acidity method lacks a standard reference (e.g., AOAC). Cite a validated protocol to ensure reproducibility. |
We have included the reference now (L140). Thanks for pointing this out. |
|
Line 149-154 (Methods - Rheological Analysis): Explain the choice of 4.5 s⁻¹ (manual scooping) and 60.8 s⁻¹ (oral mixing) with references linking to yogurt texture perception. |
We have already explained this in results part with references. L330-332. |
|
Line 182-190 (Sensory Analysis): Justify why only two samples (Y, YXL) were sensory-analyzed, excluding others (e.g., YG). Specify the testing day (e.g., day 1 or day 14). Clarify the addition of lemon flavor (10 drops) to mask umami, including concentration (drops per sample volume). Clarify why not all 49 panelists evaluated both samples (29 for Y, 20 for YXL) and why overall acceptability was not assessed, as this limits sensory insights. |
The goal was to compare the control and YXL sample only, therefore, only two samples were used for sensory analysis. The lemon flavor was used to mask umami flavor from DG, have included this in L 203. The samples were evaluated on day 1 of storage (L 201). The participants were randomly presented one of the samples to avoid bias and comparison between samples and to evaluate the liking for that sample alone. We didn’t include overall acceptability as a metric in our sensory analysis; however, to make up for that other question that was asked to participants was if this product is available in market, would you like to purchase or not. We have added more details in sensory analysis as suggested (L99-213) and also improved sensory analysis results section. |
|
Line 202 (Supplementary Table 1): Since no statistically significant differences were observed in acidity, remove the superscript letters to prevent misinterpretation. |
We have removed the superscripts from supplementary table. |
|
Line 247-249 (Discussion): The correlation between syneresis, texture, and microstructure is discussed prematurely. Consider relocating this discussion to a section following the analysis of individual results. |
We have removed this statement from section 3.2 as this has been discussed in relevant sections. |
|
Line 261-264 (Figure 2): The x-axis shows only even days (2, 4, etc.), but measurements were taken on days 1, 5, 8, and 14. Label all days (1–15) to clearly indicate measurement points. Include statistical significance annotations (e.g., letters for p < 0.05). |
We have revised the figure as suggested. |
|
Line 321-330 (Table 4): The lower viscosity of YX and YXL at 60.8 s⁻¹ (0.35–0.37 Pa.s vs. 0.64–0.67 Pa.s for Y) suggests improved oral texture, but this is not discussed (lines 371-372). Link rheological data to sensory results (lines 400-412). |
Thanks for the suggestion. We have included this in rheological analysis section. L 337-339. |
|
Line 474-496 (References): The reference list is insufficient and outdated. Include more recent studies (2023–2025), particularly those addressing novel hydrocolloids, yogurt texture, or DG/LA applications. Correct reference [12] (“V Chelikani, a.M.S.M.”) for formatting and completeness. |
We have added more recent references in the manuscript and corrected the references. |
Round 2
Reviewer 4 Report
Comments and Suggestions for Authors
Dear Authors, thank you for thoroughly addressing my comments and incorporating the suggested revisions into the manuscript. I am pleased to confirm that I am satisfied with the corrections made. Your thorough revisions have significantly strengthened the manuscript’s scientific rigor and clarity.
I appreciate your efforts and believe the manuscript is now well-suited for publication.
Author Response
Thank you for providing valuable comments for improving manuscripts. We really appreciate it.